# Coincident molecular auxeticity and negative order parameter in a liquid crystal elastomer

D. Mistry [1], S.D. Connell [1], S.L. Mickthwaite [2], P.B. Morgan [3], J.H. Clamp[4] & H.F. Gleeson [1]

Auxetic materials have negative Poisson's ratios and so expand rather than contract in one or several direction(s) perpendicular to applied extensions. The auxetics community has long sought synthetic molecular auxetics – non-porous, inherently auxetic materials which are simple to fabricate and avoid porosity-related weakening. Here, we report, synthetic molecular auxeticity for a non-porous liquid crystal elastomer. For strains above ~0.8 applied perpendicular to the liquid crystal director, the liquid crystal elastomer becomes auxetic with the maximum negative Poisson's ratio measured to date being -0.74 ± 0.03 – larger than most values seen in naturally occurring molecular auxetics. The emergence of auxeticity coincides with the liquid crystal elastomer backbone adopting a negative order parameter, $Q_B$ = -0.41 ± 0.01 – further implying negative liquid crystal ordering. The reported behaviours consistently agree with theoretical predictions from Warner and Terentjev liquid crystal elastomer theory. Our results open the door for the design of synthetic molecular auxetics.

[1] School of Physics and Astronomy, University of Leeds, Leeds LS2 9JT, UK. [2] Leeds Electron Microscopy and Spectroscopy Centre, School of Chemical and Process Engineering, University of Leeds, Leeds LS2 9JT, UK. [3] Eurolens Research, University of Manchester, Manchester M13 9PL, UK. [4] UltraVision CLPL, Commerce Way, Leighton Buzzard LU7 4RW, UK. Correspondence and requests for materials should be addressed to D.M. (email: pydam@leeds.ac.uk)

Auxetic materials have the counter-intuitive property of expanding rather than contracting perpendicular to an applied stretch, formally they have negative Poisson's ratios (PRs)[1,2]. Over the years auxetics have been shown to have enhanced mechanical properties such as energy absorption and indentation resistance over conventional materials and interesting geometrical properties such as synclastic curvature[3–5]. Consequently auxetics have potential for applications in a range of industries including sports equipment, aerospace, biomedical materials and architecture[3,5–7].

Auxeticity has been demonstrated in a variety of naturally occurring and synthetically created materials and structures. In nature, auxeticity has been observed in crystalline materials, for instance: 69% of cubic metals, α-cristobalite, and numerous of the zeolite class of materials[8–13]. In the majority of such materials, the auxetic behaviour is observed for stresses applied along specific crystal axes—i.e., the auxeticity is anisotropic[8,9,14,15]. By comparison, a much wider variety of synthetic auxetics have been reported over the years fabricated from a materials spanning metals to rubbers and with a plethora of different structures such as re-entrant honeycombs/cells, rotating units and perforated sheets[3,4,16,17]. Notably, existing synthetic materials with practically demonstrated auxetic properties are all created by carefully structuring porous geometries from positive PR materials. Crucially, it is the specific geometries of these materials, which cause the auxetic behaviour[3–5,16]. Existing synthetic auxetics are, therefore, limited by their necessary porosity, which weakens the material compared to the bulk and by the fact that such structures must be engineered, for example, by using resource-intensive additive manufacturing processes[1,18]. Recently, computer simulations have shown that non-porous, two-phase composite auxetics could be produced—potentially solving the problem of porosity-related weakening and operating via analogous deformation mechanisms of porous auxetics[17]. However, such two-phase composites are yet to be practically realised and still require manufacturing of an auxetic-inducing geometry from positive PR materials[17,19–22].

A longstanding goal for the auxetics community has been the development of a synthetic material that has intrinsic auxetic behaviour. Such molecular auxetics would avoid porosity-weakening and their very existence implies chemical tuneability[1,4,18,23–26]. However, to date no synthetic and non-porous material has been measured to have a negative Poisson's ratio[5,23,24].

Arguably polymeric liquid crystalline materials are the most promising material for displaying molecular auxeticity[26–28]. He et al.[26,27] studied a series of polydomain main-chain liquid crystal polymers and observed a strain-induced increase in the polymer inter-chain separation via X-ray diffraction. This result was taken to suggest the potential for auxetic behaviour, although to date only positive values for Poisson's ratio have been recorded in such materials[26,27].

Here, we present a synthetic molecular auxetic based on a monodomain liquid crystal elastomer (LCE). When stressed perpendicular to the alignment direction and above a critical strain of ≲0.8 the system adopts a state with a negative liquid crystal order parameter (LCOP). As the strain is further increased, the material becomes auxetic with the minimum PR measured so far being −0.74 ± 0.03 (experimental error—determined as described in the methods, n = 1). We show the auxeticity agrees with remarkable accuracy to theoretical predictions derived from the Warner and Terentjev theory of LCEs[29]. We propose that this demonstration of a synthetic molecular auxetic represents the origin of an approach to producing molecular auxetics which, in combination with the responsiveness of LCEs, could lead to exciting materials with multi-functional behaviours. Further, our results demonstrate a route for realisation of the molecular auxetic technologies that have been proposed over the years.

## Results

**NLCE properties and physical structure.** The films of mono-domain side-chain LCE were composed of 6-(4-cyano-biphenyl-4'-yloxy)hexyl acrylate (A6OCB, 34 mol%), 2-ethylhexyl acrylate (EHA, 49 mol%) and 1,4-bis-[4-(6-acryloyloxyhex-yloxy)ben-zoyloxy]-2-methylbenzene (RM82, 17 mol%) (Fig. 1a) produced as described in ref. [30], further details are given in the methods and Supplementary Fig. 1. This LCE is a modified version of a commonly used LCE, which does not display auxetic behaviour[31,32]. In addition to introducing auxeticity, the modifications reduced the glass transition temperature of the LCE from 50 °C to 14 °C and ensured a room temperature nematic phase prior to polymerisation[30]. Monodomain LCEs were produced by polymerising the monomer mixture inside liquid crystal (LC) devices coated with a uniformly oriented planar alignment agent. We have previously shown this material to be optically anisotropic (birefringence, Δn = 0.08) and also to have a highly anisotropic and non-linear stress–strain behaviour[30]. The LCE films used here were prepared with a thicknesses, d, in the range of 95–105 μm. For mechanical characterisation, strips of dimensions ~2 × 18 mm were cut with the nematic director at 89 ± 1° (s.d., n = 6) to the film long axis (Fig. 1b).

As molecular auxetics are expected to have zero porosity down to the molecular length scale it is important to assess the physical structure of our LCE across length scales. By virtue of the LCE being optically transparent (Fig. 1c) we can rule out the presence of pores from millimetre length scales, which would be

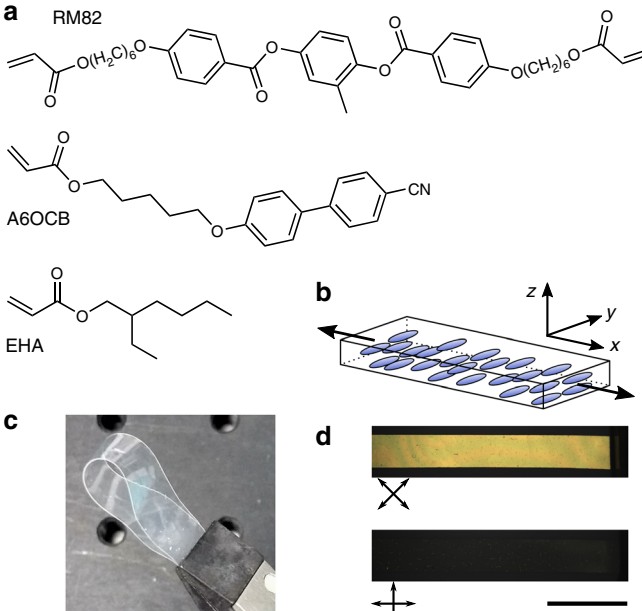

**Fig. 1** Composition and macroscopic appearance of the LCE. **a** Chemical structures of 1,4-bis-[4-(6-acryloyloxyhex-yloxy)benzoyloxy]-2-methylbenzene (RM82), 6-(4-cyano-biphenyl-4'-yloxy)hexyl acrylate (A6OCB) and 2-ethylhexyl acrylate (EHA)—the acrylate monomers used to produce the auxetic side-chain LCE. **b** Diagram of undeformed sample geometry with liquid crystal director parallel to the y-axis and perpendicular to the film long (x) axis. In this work strains ($\varepsilon_x$)/deformations ($\lambda_x$) are applied along the x-axis with the auxetic response observed along the z-axis. **c** Photograph of the final LCE showing its flexibility and high optical quality. **d** Polarising microscopy images of a unstrained LCE. Scale bar, 5 mm

individually visible, down to approximately the wavelength of visible light, which would strongly scatter light causing the material to appear milky-white[33].

The high contrast between the two polarising microscopy images shown in Fig. 2d confirms the high quality monodomain alignment and anisotropy of the LCEs produced. Moreover, the uniformity and black appearance of the sample when the polarisers are aligned with the LC director is further indication of the material homogeneity and lack of any light-scattering geometry down to sub-micron length scales.

The potential material porosity can also be directly assessed by using scanning electron microscopy (SEM) and atomic force microscopy (AFM) to study cross sections of the LCE films exposed via freeze-fracturing.

Figure 2a, b show SEM images of $yz$ cross sections according to the coordinate system shown in Fig. 1b. The homogeneous and largely featureless structure shown in Fig. 2a confirms our optical assessment of zero porosity down to sub-micron length scales. Although the micrograph with 10x greater magnification (Fig. 2b) reveals a more visible texture, nothing seen is indicative of pores and so it is evident that no porous geometry exists down to at least ~10 nm. In both Fig. 2a, b the only features visible are artefacts of the freeze-fracturing process, which in both cases confirms correct focusing of the exposed cross sections.

AFM, which is more sensitive to topographical features than SEM, allows us to probe the material structure down to the nanometre length scale. From the $yz$ cross section scans of Fig. 2c–e, two features are present: an extremely fine structure on the scale of a few nanometres; and a larger slow undulation on the scale of ~10 nm and with an amplitude of ~2 nm (comparable to the length of a liquid crystal molecule). The textures visible in Fig. 2c, d are consistent with those seen via SEM in Fig. 2b and thus confirm the LCE has zero porosity down to ~10 nm. From Fig. 2e, f we can see that the variation in the surface profile has a typical amplitude of 1 nm. As the AFM tip used had a radius of curvature of 5 nm we can further conclude that no porosity exists down to ~5 nm. From Figs. 1c, d, 2 we have assessed that there is zero detectable porosity down to length scales approaching the typical length of liquid crystalline moieties and of LCE polymer chain radii of gyration[34–38]. Consequently, the physical behaviours of the LCE we report here are inherent, bulk properties of the material itself. The question of whether porosity emerges on straining the sample is considered further below.

**Mechanical tests and auxetic behaviour**. We studied the mechanical behaviour of the LCE films using a bespoke miniature tensile rig, which simultaneously also allows observation of the polarising microscopy texture, thus providing an insight into the LCOP[30]. The films were extended along the $x$-axis (as illustrated in Fig. 1b) in steps at various extension speeds, based on the percentage increase in sample length per minute relative to the unstrained sample length, $L_0$, and at various temperatures as summarised in Table 1. For each mechanical test (each performed on a different sample) the strains $\epsilon_x$ and either $\epsilon_y$ or $\epsilon_z$ were recorded from photographs of the sample. In all but one test the $xy$ plane was observed and the strains $\epsilon_x$ and $\epsilon_y$ measured; from these data $\epsilon_z$ could be inferred using the constant volume condition (Eq. 2, methods). From the pairs of strains, $\epsilon_x$ and $\epsilon_z$ or $\epsilon_x$ and $\epsilon_y$, the instantaneous PRs, $\nu_{xz}$ or $\nu_{xy}$, respectively, were calculated according to the method described by Alderson et al.[39] and Smith et al.[40] (see equations 3–5, methods).

Figure 3 demonstrates how the deformations of the LCE perpendicular to the director are volume conserving and do not induce the formation of any porous microstructure. The two $\epsilon_z - \epsilon_x$ curves shown in Fig. 3a were calculated from separate tests (both under test conditions I) on separate samples where the strains $\epsilon_z$ were measured from direct observation of the $xz$ plane as well as being inferred through application of the constant volume condition to the measured strains $\epsilon_x$ and $\epsilon_y$. The

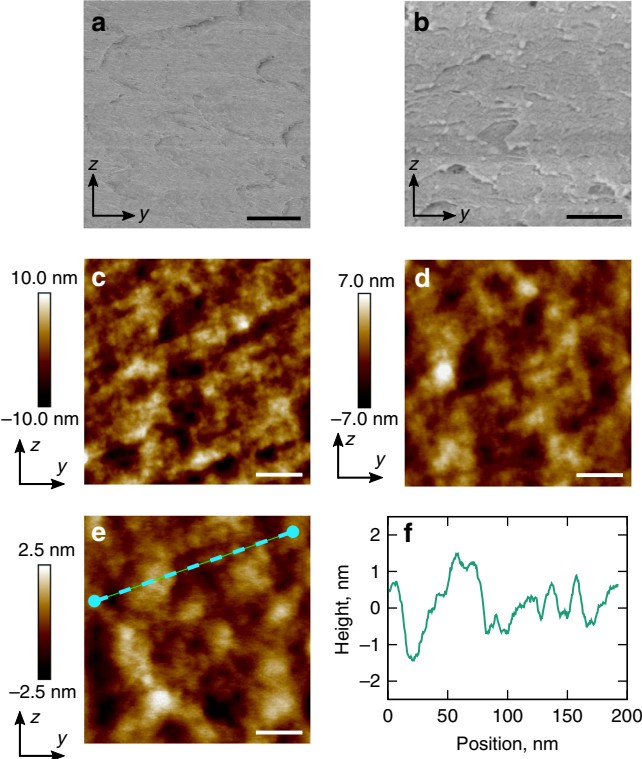

**Fig. 2** Micro- and nano-scopic structure of the LCE. **a, b** Cryo-SEM micrographs (scale bars, 2.5 μm and 250 nm, respectively) and **c–e** AFM height maps (scale bars, 200, 100 and 40 nm, respectively) show the structures present from molecular to microscopic length scales. Vertical colour bars accompanying AFM height maps represent height in nm. Axes used in **a–e** correspond to the coordinate set from Fig. 1b. **f** Profile of dashed line drawn across **e**

| Table 1 Testing parameters and PR at the maximum extension | | | |
|---|---|---|---|
| Test | Extension speed (% $L_0$ min$^{-1}$) | Temperature ( ±1 °C) | Critical strain ($\varepsilon_c$) emergence of auxeticity | PR at maximum extension |
| I | 16 | 28 | 1.02 ± 0.02 | -0.50 ± 0.10 |
| II | 7.5 | 28 | 0.79 ± 0.03 | -0.59 ± 0.08 |
| III | 1.0 | 24 | 0.80 ± 0.01 | -0.56 ± 0.09 |
| IV | 0.71 | 23 | 0.82 ± 0.02 | -0.74 ± 0.03 |

Error on temperature readings based on range of readings from three thermocouples distributed around the samples. Errors on $\varepsilon_c$ indicate measurement error on strains measured at the emergence of auxeticity for each test ($n = 1$). Errors for PR at maximum extension described in the methods section ($n = 1$)

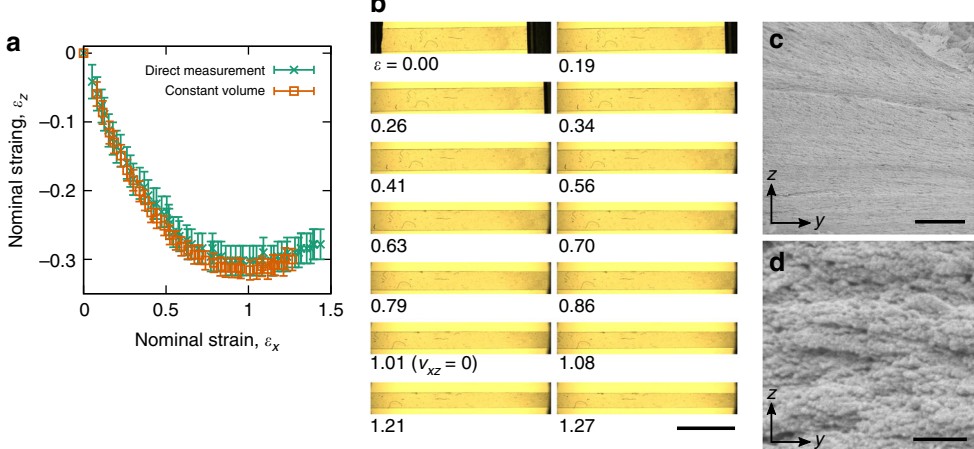

**Fig. 3** Material volume conservation and microstructure with strain. For samples tested under condition **I**: **a** $z$-axis strains, $\epsilon_z$ in response to imposed $x$-axis strains, $\epsilon_x$, measured via direct observation of the LCE $xz$ plane (green crosses) and through application of the constant volume condition to strain measurements of the $xy$ plane (orange boxes). Errors are measurement errors ($n = 1$). **b** Photographs of the $xy$ plane of the LCE showing consistent transparent and non-scattering, or stress-whitened, appearance. Scale bar, 5 mm. **c**, **d** Cryo-SEM images of a sample strained to the auxetic regime and freeze fractured to expose a $yz$ cross section. Scale bars, 2.5 μm and 250 nm, respectively

consistency between the direct and inferred measurements of $\epsilon_z$ demonstrates the validity of the constant volume condition for the LCE. As such, we can conclude that the LCE density is constant with strain—a result to be expected for non-porous elastomers that are typified by possessing bulk moduli, which are several orders of magnitude greater than their shear moduli[29,41–43]. Moreover, the volume conserving nature of the material rules out the formation of wrinkles across the width of the films.

Figure 3b shows photographs of a sample illuminated via transmitted light at various strain steps of a deformation. As the transparency and apparent brightness of the sample remains constant, we can deduce that no light-scattering ($\lesssim 400$nm) textures (such as cavitation-induced pores or strain-induced crystals) are forming within the material for the range of strains tested[32,44]. Fig. 3c, d shows a cryo-SEM freeze fracture of the $yz$ plane of a LCE sample deformed into the auxetic region (cryogenic freezing prevented relaxation of the film following the fracture—see Methods). As one might expect for a highly strained sample, the apparent texture of the exposed surface differs compared to the surface seen in Fig. 2a, b for the unstrained sample. In agreement with the conclusions drawn from Fig. 3b, c also shows no evidence of cavitation or strain-induce-crystallisation features on sub-micron length scales. Figure 3d further confirms that in the auxetic region, the LCE remains non-porous as the only features visible are ~20 nm features protruding out of, and into the exposed surface. These conclusions drawn from Fig. 3b–d are in agreement with Fig. 3a as if any strain-induced pores or crystals were to develop within the sample then material would not deform at constant volume as is clearly shown in Fig. 3a[45].

Figure 4a shows for conditions **I–IV** strains, $\epsilon_z$, calculated via the constant volume assumption using strains measured in the $xy$ plane (see Methods). Like the curves in Fig. 3a, in each case the strain $\epsilon_z$ approaches a minimum before increasing (Table 1 and Fig. 3a). Auxetic behaviour is demonstrated in the regions of further increasing $\epsilon_z$ where the sample is growing thicker in the direction transverse to the increasing applied strain, $\epsilon_x$. Figure 4b shows for each test the instantaneous PR, $\nu_{xz}$, against nominal strain $\epsilon_x$. The curves confirm the emergence of auxetic behaviour above a critical strain, $\epsilon_c$, above which $\nu_{xz}$ becomes negative. The value of $\epsilon_c$ for each test shows a dependence on the sample

temperature and on the speed at which it is extended (Table 1), most likely a result of the different conditions allowing different levels of stress relaxation between successive extensions. However, if the behaviour of $\nu_{xz}$ for each case is considered with respect to $\epsilon_c$, then we can see that the magnitude of the auxetic response is in fact infact largely identical and hence independent of extension speed and temperature (Fig. 4b). In each case shown in Fig. 4b, c, $\nu_{xz}$ for the most part monotonically decreases with strain. A minimum value of $\nu_{xz} = -0.74 \pm 0.03$ was recorded in test **IV** which was strained by the largest factor (~1.7) above $\epsilon_c$.

Figure 4d and e show corresponding plots of the $\epsilon_x$-$\epsilon_y$ strains measured and instantaneous PRs $\nu_{xz}$ calculated for tests **I–IV**. Comparing these graphs with Fig. 4a, b, it is clear that the deformation behaviour of the LCE in the $xy$ and $xz$ planes differs significantly—i.e., the deformation is highly anisotropic. In contrast to $\nu_{xz}$, in each case $\nu_{xy}$ is initially near-zero and increases with strain. The anisotropy between $\nu_{xy}$ and $\nu_{xz}$ for our material should be expected for two reasons. Firstly, monodomain LCEs are well known to have inherent mechanical anisotropy[28,29,46,47]. Secondly, simultaneous volume conservation and Poisson's ratio isotropy is only possible in the specific case of $\nu_{xy} = \nu_{xz} = 0.5$, for which there is no auxetic behaviour[48]. These conditions are formalised mathematically by the condition (for volume conserving materials) $\nu_{xy} + \nu_{xz} = 1$, which is derived in the methods.

**Negative liquid crystal order parameter (LCOP).** Figure 5 shows polarising microscopy images of the LCE deformed under test conditions **I**. In the unstrained state the sample has an optical retardance $\Gamma = \Delta n \times d \approx 8000$ nm (~14th order). The progression to birefringence colours of increasing saturation indicates a decrease in retardance (to ~1st–3rd order between $\epsilon_x = 0.87$ and $\epsilon_x = 1.06$) that cannot be accounted for by the (initially) reducing sample thickness (Fig. 4a)[30]. Such behaviour is associated with a reducing LCOP within the $xy$ plane. The black appearance at ($\epsilon_x = 1.14$) exists for all sample rotations with respect to the crossed polarisers (Supplementary Fig. 2) meaning there is zero birefringence and hence zero LC ordering *within the $xy$ plane*—i.e., the LCE is isotropic within the $xy$ plane. We have previously deduced that such an observation corresponds to a state of negative LCOP[30]. Below we confirm this behaviour by

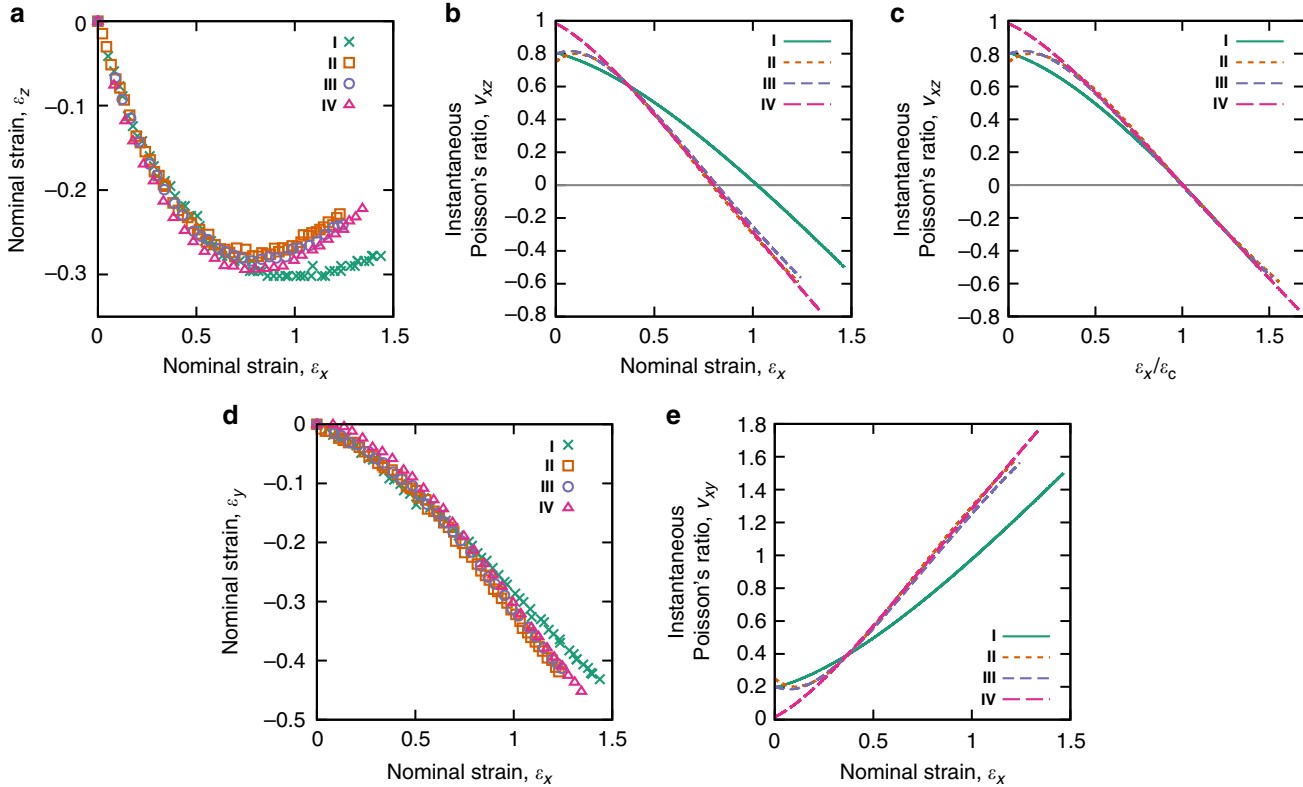

**Fig. 4** Measured strains and instantaneous Poisson's ratio behaviour. **a** For each of the various samples deformed under conditions **I–IV** the strain $\varepsilon_z$ initially decreases. Beyond the minimum the behaviour becomes auxetic. **b** Strain-dependent Poisson's ratios extracted from a shows that $\nu_{xy}$ in general monotonically decreases and becomes negative above a critical strain, $\varepsilon_c$, specific to each test condition. **c** The instantaneous Poisson ratios plotted relative $\varepsilon_c$ showing that the behaviour of $\nu_{xz}$ in all cases are largely identical. **d** The measured principle strains in the $xy$ plane and **e** the calculated instantaneous PRs, $\nu_{xy}$, for all test conditions **I–IV**. In each case $\nu_{xy}$ begins close to zero and increases with imposed strain, $\varepsilon_x$

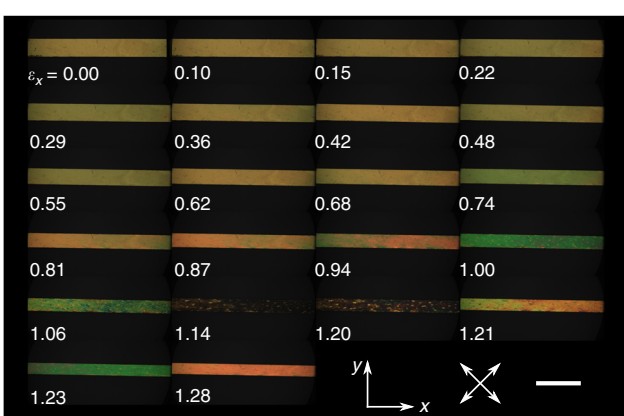

**Fig. 5** Negative LCOP order from polarising microscopy. Polarising microscopy textures at each strain step of test **I**. The birefringence colours indicate the retardance initially decreases, becoming zero at $\varepsilon_x = 1.14$, before increasing again. Scale bar, 5 mm

applying the theory of LCEs pioneered by Warner and Terentjev (W&T)[29]. Beyond $\varepsilon_x \sim 1.20$, birefringence colours re-emerge and hence the in-plane LCOP increases, but with the director now parallel to the stress axis[30].

The results presented here can be tested against the theory of W&T, which speculated about the possibility of auxetic behaviour in aligned LCEs. The theoretical suggestion has, however, never

previously been observed in experiments or simulations. Under the Gaussian theory of elasticity, the polymer chain for an isotropic elastomer can be modelled as a random walk with effective steps of length $l$. On average, a polymer chain will, therefore, adopt a spherical conformation. However, in a nematic LCE the anisotropic ordering of the mesogenic groups is templated onto the polymer chain conformation resulting in an anisotropic and (most generally) biaxially ellipsoidal shape (Supplementary Fig. 3)[29]. The effective step of the polymer chain random walk is, therefore, anisotropic and is described by the effective step length tensor, $\underline{l}$, which equals $\mathrm{Diag}(l_1, l_2, l_3)$ in the principal frame[29]. The relationship between the effective step lengths and the polymer conformation is shown in Supplementary Fig. 3. Crucially, the coupling between the side-chain LC moieties and polymer backbone means the LCOP tensor and $\underline{l}$ have a common symmetry[29]. As typically the LCOP has uniaxial symmetry, the effective step length tensor is given by $l_1 = l_\parallel$ and $l_2 = l_3 = l_\perp$ where the unique axis is aligned parallel with the nematic liquid crystal director. The ratio $r = l_\parallel / l_\perp$ is known as the step length anisotropy and characterises the anisotropy of the polymer backbone for uniaxial systems.

In their theory of LCEs, W&T speculated that an auxetic response of LCEs may be observed to begin at a deformation given by $(\lambda_x, \lambda_z) = \left(r_0^{1/3}, r_0^{-1/6}\right)$, where $r_0$ is the step length anisotropy for the unstrained LCE and the deformations $\lambda_i$ $(= \varepsilon_i + 1)$, are components of the deformation gradient tensor, $\underline{\underline{\lambda}}$. Considering data for test **I** (chosen as we show corresponding polarising microscopy images in Fig. 5) from Fig. 4a, we calculate

using the critical strains of $(\epsilon_x, \epsilon_z) = (1.02 \pm 0.02, -0.30 \pm 0.02)$ values of $r_0 = 8.2 \pm 0.3$ and $r_0 = 8.5 \pm 0.7$, which are comfortably self-consistent.

Further independent calculations of $r_0$ can be made by applying W&T theory (see methods) to the strain at which we have observed zero birefringence (Fig. 5). Our ability to use this theory stems from the following deductions about the symmetry of $\underline{\underline{l}}$ for the LCE in the unstrained state and at the black state seen in Fig. 5. For the unstrained LCE, the LCOP has nematic (uniaxial) symmetry and the director lies along the $y$-axis—therefore meaning $\underline{\underline{l}} = \underline{\underline{l}}^0 = \mathrm{Diag}(l_\perp^0, l_\parallel^0, l_\perp^0)$. As in the black state the LCOP is deduced to be isotropic within the $xy$ plane, the effective step lengths $l_1$ and $l_2$ must in this state be equal—therefore meaning $\underline{\underline{l}} = \underline{\underline{l}}' = \mathrm{Diag}(l'_\perp, l'_\perp, l'_\parallel)$, which has principal axes parallel with those of $\underline{\underline{l}}^0$. Further details of how these deductions are used to calculate values for $r$ in the unstrained and black states are given in the methods.

From Fig. 5 we can deduce for test **I** that at the state of a negative LCOP $\lambda_x = 2.15 \pm 0.05$ and $\lambda_y = 0.67 \pm 0.05$ (measurement errors, $n = 1$). By Inserting these values in to Eq. 9 (methods) and solving, we find $r_0 = 10.2 \pm 1.6$ and $r' = 0.103 \pm 0.015$ (propagation of measurement errors through Eq. 9, $n = 1$). This calculation of $r' < 1$ corresponds to the system adopting a uniaxial oblate polymer conformation. Previously we have determined values of $r_0 = 9.3$ and $r_0 = 3.8$ from opto-mechanical and thermal tests, respectively[11]. At the time, comparisons of these values to those from LCEs of similar chemistries led us to conclude that the latter value was more likely to be accurate[30]. However, the several independent calculations of $r_0 \sim 9$ presented here now leads us to believe that this value is actually most likely to be correct. The self-consistency of values presented here demonstrates that W&T theory describes the physical behaviour of our material well—hinting that the auxetic response of future materials could be tuned by controlling the magnitude of $r_0$.

Using the calculated values of $r$ we can also extract values for the polymer chain backbone order parameter, $Q_b$, using the equation[49]

$$r = \frac{1 + 2Q_b}{1 - Q_b} \qquad Q_b = \frac{r - 1}{r + 2} \qquad (1)$$

which gives $Q_b^0 = 0.74 \pm 0.03$ and $Q_b' = -0.41 \pm 0.01$ (propagation of measurement errors through Eq. 1, $n = 1$) which, given the shared symmetry between the LCOP and $\underline{\underline{l}}$ tensors, are consistent with our present and previous deductions of the LCOP symmetry[30].

## Discussion

In Fig. 6 we bring together our results to illustrate the deduced relationships between: the macroscopic deformations; the polymer conformation shape; and the LC ordering with projections onto the $xy$, $yz$ and $zx$ planes. The illustration of Fig. 6 shows that the point corresponding to $v_{xz} = 0$ is a point of symmetry for the deformation behaviour of the LCE studied here—suggesting that the transition to the auxetic regime is required by symmetry.

The auxetic behaviour in the LCE we present appears to be an inherent material property and not the consequence of a porous structure. Hence, this LCE is the first example of a synthetic molecular auxetic. This observation is particularly significant as LC polymers and LCEs have arguably been considered as the most promising type of synthetic material for displaying molecular auxeticity, but a negative Poisson's ratio has never previously been measured[23,26,27,29,50,51]. The maximum magnitude of negative PR observed in this system, $-0.74 \pm 0.03$, is larger

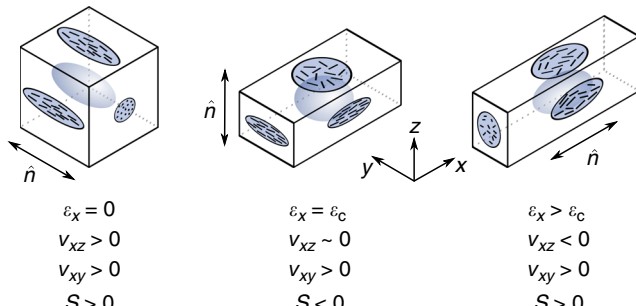

**Fig. 6** Relationships between LC order and the strain-dependent Poisson's ratios. Model of the deformation described by: the relationships between the sample geometry (outline box), polymer conformation shape (enclosed ellipsoidal shapes) and LCOP (denoted as $S$) projected on each plane (rod arrangements). At the critical strain, $\epsilon_c$, the symmetry of the LC ordering corresponds to a negative order parameter with the director (symbolised by vectors-n) lying parallel to the $z$-axis

than most values seen in naturally occurring molecular auxetics such as α-cristobalite ($\nu \sim -0.5$), cubic metals (broad range of negative $\nu$ calculated between $0 \lesssim \nu \lesssim -0.8$) and zeolites (broad range negative $\nu$ calculated between $0 \lesssim \nu \lesssim -1.2$)[8,9,14]. While the main focus of this paper has been the auxetic behaviour of the reported LCE, it is also interesting to note that at low strain $\nu_{xy} \sim 0$ and $\nu_{xz} \sim 1$—i.e., the majority of the deformation occurs in the thickness direction of the film. This property could find use in substrates for soft electronic devices where, for instance, wires lying parallel to the LC director would be minimally affected by the substrate being moderately stretched perpendicular to the director.

In this study, the close coincidence of $\epsilon_c$ with the negative LCOP state suggests the emergence of auxeticity and the state of negative liquid crystalline ordering are intrinsically linked. However, it remains to be seen whether the observed coincidences are common features for auxetic LCEs of different chemistries. Certainly, further experimental studies investigating the anisotropic and strain-dependent Poisson's ratio for a variety of different LCE chemistries is required to understand the critical chemical aspects responsible for the auxetic behaviour. Moreover, coupling these experiments with computational and theoretical studies will allow a molecular picture of the auxetic response to be built. From this foundation one can envisage being able to design LCE-based auxetic materials with properties such as auxeticity from zero strain. Based on the relationships derived from W&T, we propose that the first step toward developing this desired molecular understanding would be to use well known chemical modifications to moderate the value of $r_0$ and hence tune the auxetic response.

Looking forward, LCEs offer an exciting prospect for coupling the photo-, thermal-, and chemo-actuation behaviours with auxeticity[29,52–54]. Additionally, one can readily envisage the photo-switching of the auxetic behaviour through the inclusion of mesogenic monomers containing azo-benzene groups[54–57]. Moreover, as to the best of our knowledge a transparent auxetic material has never been previously observed, the presented LCE opens the future possibility of optical-auxetic sensors and devices.

## Methods

**Preparation of monodomain side-chain LCEs**. LCE precursors were prepared from the following acrylate monomers: 6-(4-cyano-biphenyl-4'-yloxy)hexyl acrylate (A6OCB, 15 mol%), 2-Ethylhexyl acrylate (EHA, 21 mol%), 1,4-Bis-[4-[6-acryloyloxyhex-yloxy)benzoyloxy]-2-methylbenzene (RM82, 7 mol%) (Fig. 1a), and the liquid crystal 4'- hexyloxybiphenyl (6OCB, 56 mol%) and photoinitiator

methyl benzoylformate (MBF, 1.6 mol%) (Supplementary Fig. 1)[30]. Use of 6OCB ensured the LCE precursor had a broad nematic phase enabling alignment of the liquid crystal at room temperature prior to polymerisation[30]. Using a balance of 0.3 mg accuracy, quantities of A6OCB, 6OCB and RM82 were first dispensed into a glass vial and heated to 120 °C in order to melt each component and allow them to mix together. The mixture was then cooled to ambient temperature and the EHA and MBF were added using an Eppendorf pipette. The vial cap was replaced and the precursor mixed for 5 min on a hotplate stirrer set to 40 °C and 80 rpm. At this temperature the precursor was in its isotropic phase ($T_{NI} = 36$ °C) and evaporation of the volatile EHA was minimised. The precursor was then capillary filled at 40 °C into LC devices prepared from one glass and one Melinex® substrate, coated with a rubbed poly vinyl(alcohol) alignment layer and separated using 75 μm strips of spacer film coated with an adhesive. The devices were allowed to cool to ambient temperature and were left for 20 min to allow alignment of the liquid crystal. LCEs were then polymerised using a 2.5 mW cm$^{-2}$ fluorescent UV light source for 2 h to ensure complete polymerisation. Following polymerisation the Melinex® substrates were peeled away and the LCE removed from the device. 6OCB and any unreacted MBF were then washed from the LCE by placing the film in a methanol:DCM 70:25 solution for 2 h. The sample was then rinsed in methanol and hung to dry at 50 °C overnight resulting in the final LCEs.

**Cryo-SEM.** Samples of LCE were studied via Cryo-SEM using a Quorum Technologies PP3010 Cryo-FIB/SEM preparation system and a FEI Helios G4 CX SEM. LCE sample were placed into precooled cryo-SEM shuttles, which were then placed into liquid nitrogen. The shuttle and liquid nitrogen were placed under vacuum and pumped until just before the nitrogen became solid. The sample and shuttle were then pulled into a vacuum pot and transferred into the preparation chamber where the sample was kept at –140 °C under high vacuum. The LCEs were then fractured with a cooled knife and sublimed for 3 min at –90 °C to remove any frost on the sample before being returned to –140 °C. To avoid charging effects, the exposed $yz$ cross sections (Fig. 1b) were then sputter coated with Platinum (5 mA for 45 s). The sample shuttle was then transferred into the SEM and imaged using a 1 kV accelerating voltage and a 0.10 nA beam current. Minor brightness and contrast adjustments were globally applied to the captured images using ImageJ (US NIH)[58].

For imaging the LCE structure in the auxetic regime, the LCE was stretched using a miniature manual stretching rig. By viewing the LCE via crossed polarisers we could determine when then auxetic regime had been reached by observing the progression of birefringence colours through the sequence shown in Fig. 5. Once in the auxetic regime, the manual stretcher and LCE were placed in a pot containing liquid nitrogen and once the nitrogen had stopped boiling, the stretched sample was removed (while kept in liquid nitrogen) from the manual stretcher. The sample was then placed in the precooled cryo-SEM shuttle and fracture performed as described above. At all stages of the experiment after the LCE was stretched, the sample was kept at cryogenic temperatures—far below the LCE's glass transition temperature ($+ 14.0 \pm 1.0$ °C, error given is a combination of instrument and fitting errors)—in order to prevent the imposed strain relaxing away.

**Atomic force microscopy (AFM).** $yz$ cross-sectional samples (coordinate system shown in Fig. 1b) were prepared by encasing samples of LCE within a two-part epoxy glue and then freeze-fracturing the exposed LCE. AFM images were acquired using a Bruker Dimension FastScan-Bio, using Bruker FastScan A probes, in air tapping mode at a frequency of 1.4 MHz. Images were acquired at a line rate of approximately 4 Hz at 1024 pixel resolution, then processed with a simple low order line flattening in Bruker Nanoscope Analysis v1.9.

**Mechanical testing.** The specification of the bespoke equipment used to measure the sample deformations and polarising microscopy textures is described at length in ref. [30]. and accompanying supplementary information[30]. The equipment comprises a microtensile rig, which allows the sample under test to be observed via transmitted or reflected white light and polarising microscopy. The actuators are capable of extending the sample with a minimum step size of 0.5 mm. The sample deformations are observed using a camera with a 4.2 MPx (2048 × 2048 px) sensor and a lens with a magnification, which can be varied between x0.7 and x13. For samples tested under each conditions I–IV the $xy$ plane of each sample was photographed (x0.7 magnification lens used with field of view of 16 × 16 mm) at each strain step. From transmitted white light images taken at each strain step the strain $\epsilon_x$ was measured using the relative separation of features of the samples (initial separation of around 250px) and the strain $\epsilon_y$ was measured from the changing sample width. The position of tracked features was typically measured from photographs with a 2px accuracy. Under the constant volume assumption, an initial volume element of dimensions $l_0 \times w_0 \times t_0$ will maintain a constant volume as the element is deformed to dimensions $l \times w \times t$. The strain, $\epsilon_z$, in the thickness direction can, therefore, be determined via

$$\epsilon_z = \frac{t}{t_0} - 1 = \frac{l_0}{l} \times \frac{w_0}{w} - 1 \quad (2)$$

The validity of the use of the constant volume assumption was confirmed by directly measuring the change in thickness of a second sample tested under

conditions I and comparing the measured deformation with the data from above method (Fig. 3a). For this test, the sample edge ($xz$ plane) was illuminated via reflected light and the lens magnification was increased by a factor of 13. The camera resolution in this mode was 3 μm. As the initial sample thicknesses was typically 100 μm, the thickness could be measured with sufficient precision to confirm the validity of the constant volume assumption for the material investigated. From photographs taken, the thickness was determined by a python script which at each strain step automatically measured the thickness in each column of pixels. The median value and median absolute deviation were taken as the sample thickness and error respectively, from which the strain $\epsilon_z$ was directly determined.

As the deformations of the sample were highly non-linear, it was important to report the instantaneous PR in order to characterise how the LCE deformed with strain. The instantaneous PR can be calculated using the method employed by Alderson et al.[39] and Smith et al.[40], which is described as follows.

For a sample of original dimension $L_0^i$ deformed along the $i^{th}$ axis from $L^i$ to $L^{i'} = L^i + \delta L$ (where $\delta L$ is small), the instantaneous strain is given by

$$\epsilon_i^{INST} = \frac{L^{i'} - L^i}{L^i} = \frac{L^{i'}/L_0^i - L^i/L_0^i}{L^i/L_0^i} = \frac{\delta\lambda_i}{\lambda_i} \quad (3)$$

where $\lambda_i = \epsilon_i + 1$ is the sample deformation (ratio of the sample's current length to initial length). In the differential limit the instantaneous strain is given by

$$\epsilon_i^{INST} = d\ln(\epsilon_i + 1) = d\epsilon_i^t \quad (4)$$

where $\epsilon_i^t$ is the true strain of the sample deformation. The instantaneous PR, $\nu_{ij}$, can, therefore, be calculated from the negative ratio of instantaneous deformations perpendicular and parallel to the an applied strain

$$\nu_{ij} = -\frac{d\ln\lambda_j}{d\ln\lambda_i} = -\frac{d\epsilon_j^t}{d\epsilon_i^t}, \quad (5)$$

i.e., the negative gradient of the true strains perpendicular parallel to the applied strain axis. For our calculations we fitted fourth-order polynomials to true strain plots of the measured deformations. The fitted functions were restricted to pass through the absolute point (0,0). The negative gradient of these curves gave the instantaneous PR as a function of strain (plotted in Fig. 4b against nominal strain). In order to estimate the errors on the critical strain at which auxeticity emerges ($\epsilon_c$) and the instantaneous PR at maximum extension, the above procedure was replicated using curves fitted to the upper and lower error bounds on the measured strain data. Supplementary Fig. 4 shows and example plot of the curves fitted to $\epsilon_x^t$-$\epsilon_z^t$ for one of the samples tested under conditions I. The error for $\epsilon_c$ was determined by the upper and lower limits for the strain at which the instantaneous PR equals zero.

From Eq. 5 and the constant volume assumption, the following link between the instantaneous PRs $\nu_{xz}$ and $\nu_{xy}$ is easily deduced

$$\nu_{xy} + \nu_{xz} = 1 \quad (6)$$

**Theory of negative order parameter state.** The elastic free energy, $F_{el}$, of a LCE is given by the Warner and Terentjev trace formula[29],

$$F_{el} \propto \mathrm{Tr}\left(\underline{\underline{l}}^0 \cdot \underline{\underline{\lambda}} \cdot \underline{\underline{l}} \cdot \underline{\underline{\lambda}}\right) + \ln\left(\frac{\mathrm{Det}\left(\underline{\underline{l}}\right)}{\mathrm{Det}\left(\underline{\underline{l}}^0\right)}\right) \quad (7)$$

In this case we assume shear-free deformations, i.e., $\underline{\underline{\lambda}} = \mathrm{Diag}(\lambda_x, \lambda_y, \lambda_z)$ which we can simplify using the constant volume condition of $\bar{\lambda}_x = 1/\lambda_y\lambda_z$. Following ref. [59] and minimising $F_{el}$ with respect to $\lambda_y$ and $\lambda_x$ gives rise to predictions for the physical state of the LCE at $\epsilon_x \approx 1.15$. From this, the following equations are derived for the deformation, $\lambda_y$, along the $y$ axis, and the elastic free energy, $F_{el}$, for a general deformed state of $\underline{\underline{l}} = \mathrm{Diag}(l_1, l_2, l_3)$ which, like here, has principle axes parallel to those of $\underline{\underline{l}}^0$:

$$\lambda_y = \frac{1}{\sqrt{\lambda_x}}\left(\frac{l_y l_\perp^0}{l_z l_\parallel^0}\right) \text{ and } F_{el} \propto \lambda_x^2 \frac{l_\perp^0}{l_x} + \frac{2}{\lambda_x}\sqrt{\frac{l_\perp^0 l_\parallel^0}{l_y l_z}} \quad (8)$$

where $\lambda_x$ is the deformation along the extension direction.

By substituting $l_1 = l_2 = l_\perp'$, and $l_3 = l_\parallel'$ into the above and minimising $F_{el}$ with respect to $\lambda_x$ we arrive at the following relationships:

$$\lambda_x^6 = \frac{r_0}{r'} \text{ and } \lambda_y^6 = \frac{1}{r'r_0^2} \quad (9)$$

where $r_0 = l_\parallel^0/l_\perp^0$ and $r' = l_\parallel'/l_\perp'$ are the step length anisotropies of the unstrained and strained states respectively. Note, the above does not placed any constraint on the value of $r'$ with respect to unity.

## Data availability

Source data files, plotting files, original AFM and SEM images and figure generation files are available at [https://doi.org/10.5518/462]. Raw data can be obtained from the authors upon reasonable request[59].

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

## Acknowledgements

D. Mistry thanks UltraVision CLPL and the EPSRC for a CASE Ph.D studentship and the Royal Commission for the Exhibition of 1851 for an Industrial Fellowship. We thank T. Haynes and P. Thornton for building equipment, Mark Warner for useful discussions and the Leeds Electron Microscopy And Spectroscopy centre (LEMAS).

## Author contributions

D.M., H.F.G, J.C., and P.B.M. conceived the research and designed the material. D.M. produced the material and performed the optical and mechanical characterisation. S.C. performed the AFM and analysed the results. S.L.M performed the cryo-SEM. D.M. analysed the optical, mechanical and opto-mechanical results. D.M. and H.F.G prepared the manuscript. All authors reviewed the manuscript.

## Additional information

**Competing interests:** D.M., H.F.G., and P.B.M. have filed a patent relating to auxetic LCEs. All remaining authors declare no competing interests.

