## [Peer Review File · Nature Communications]

Reviewers' comments:

Reviewer #1 (Remarks to the Author):

This paper is of significant importance if its claims are fully substantiated. The results are novel and largely original however some further work needs to be done for the work to be completely convincing.

Detailed comments:

Line 16

The authors have failed to reference the several papers on intrinsic auxeticity found in specific zeolites. This needs to be addressed.

Line 42

"A molecular auxetic is expected to have.....across all length scales" is highly contentious. More accurately I would suggest:

"A molecular auxetic is expected to have.....down to molecular length scales" – which is, in fact, what the data show (line 46).

Measurements of porosity are made. However the samples may only reveal this porosity under strain – in the negative Poisson's ratio region. The fact that there is no porosity when unstrained is not conclusive evidence that the mechanism is actually at the molecular level, unless the tests are also made on samples held under strain in the auxetic region.

Line 58 etc and Lines 182

I have many concerns about the methodology for measuring and calculating Poisson's ratio. The authors make no reference to the literature on this aspect where it is essential to distinguish between small strain and large strain and to use "true strain" at large strains, without reference to the original length.

Also, using a constant volume assumption when changes in Poisson's ratio are all about the volume changing is very dubious. This is also related the issue of whether the material is actually highly anisotropic and whether the formulae used are not entirely valid in the anisotropic case.

Whilst there does indeed appear to be a negative Poisson's ratio effect at large strains there needs to be greater clarity on the method of calculation used, reference to earlier works in this area and clarity on the assumptions made about constant volume and anisotropy before the values quoted can be accepted.

Another important issue is the number of samples tested. I found this difficult to ascertain. Is this simply one sample tested at four different strain rates. Ideally one would wish to see at least 5-10 different samples tested and also some estimate of error bars on the values of Poisson's ratio quoted.

Reviewer #2 (Remarks to the Author):

The authors show that in their investigation of a liquid crystal elastomer, it was found that the auxetic behaviour is inherent rather than due to porous structure. The work is not novel, as this has already been taken by Griffin et al 2 decades ago. The specific method employed, however, is new, as does the choice of chemical system.

Major comments

1) Perusal to figure 3 suggests that the authors adopt the incremental strains, rather than overall strain, for their definition of Poisson's ratio. For example, fig 3a and d show that the strain in z

direction in negative throughout the detention in x direction , but fig 3 b and c show that the PR is negative beyond certain range of stretch, which presumably indicated by the upward change in the z strain. As such, the plots of PR in fig 3 b and c should be clarified as "incremental PR".

2) the quantities I_1 , I_2 , I_3 as well as λ_{ex} , λ_u , λ_z are not clear. May I suggest that these be explicitly displayed and defined on a diagram to prevent any ambiguity and possible misunderstanding that may arise for the readers.

Minor comments

a) where is equation (1)?

b) in the first page, the authors missed out some very recent reviews of auxetic systems. These should be incorporated in the early part of the paper for the sake of completeness. The reviews are listed below.

X. Ren et al, *Smart Materials and Structures* 27, 023001 (2018)

R. Lakes, *Annual Review of Materials Research* 47, 63-81 (2017)

T.C. Lim, *Physica Status Solidi RRL* 11, 1600440 (2017)

Reviewer #3 (Remarks to the Author):

In this study, Mistry et al reported the auxetic property in a synthetic monodomain liquid crystal elastomer consisting of acrylate monomers. A series of characterization were performed to understand the material morphology, mechanical properties and auxetic property. The results show a minimum Poisson's ratios of -0.8 was achieved when material strain was elongated more than 0.8. Different from previously reported auxetic materials with porous structure, the authors claim the first example of a synthetic molecular auxetic in this study. The reviewer feel the topic is interesting and may bring broad reading interest from the scientific community. The following concerns and issues should be considered to further improve the quality of the manuscript before it could be accepted for publication.

1. To further demonstrate the hypothesis in the current design concept, chemical functionalization of the polymer backbone or side chain in relation to the increase of r_0 value should have been done. The data presented in current study is based on a single sample tested under different conditions, and the material has been previously reported in *Soft Matter*, 2018,14, 1301-1310. The reviewer feels it is difficult to do comparison and structure-property correlation investigation from different aspects of the materials.

2. In addition, to prove the strategy in the material design is versatile, proper physical modifications to achieve an increased magnitude of liquid crystal order parameter so as to obtain a greater negative PR should have been also considered and correlated with underlying mechanism of polymer chain motion in molecular level.

3. In addition to the morphology studies by SEM and AFM, as well as relationships between LC order and auxetic behaviour, the authors are suggested to provide more evidence on in-depth study and insightful discussion in relation to the underlying mechanism.

4. Authors are also suggested to demonstrate at least a niche application by using the unique molecular auxetics and its physical properties/functional behaviours.

Devesh Mistry
School of Physics and Astronomy
University of Leeds
Leeds, L S2 9JT
pydam@leeds.ac.uk
15th August 2018

Dear Referee,

Firstly thank you very much for your constructive comments on our manuscript titled “*Coincident Molecular Auxeticity and Negative Order Parameter in a Liquid Crystal Elastomer*”. Following your comments we have acquired additional data and have substantially revised our manuscript to address your comments. In addition, substantial changes to the abstract and introduction have now been made to better reflect the style of articles published by *Nature Communications*. We hope that you agree that our manuscript addresses your comments and is now strong enough for publication. Below we provide a point-by-point description of how we have addressed your individual comments.

Reviewer 1 comments:

This paper is of significant importance if its claims are fully substantiated. The results are novel and largely original however some further work needs to be done for the work to be completely convincing.

Detailed comments:

- 1) Line 16 - The authors have failed to reference the several papers on intrinsic auxeticity found in specific zeolites. This needs to be addressed.

Reference to zeolites has been made in the introduction and relevant papers cited

- 2) Line 42 - “A molecular auxetic is expected to have.....across all length scales” is highly contentious. More accurately I would suggest: “A molecular auxetic is expected to have.....down to molecular length scales” – which is, in fact, what the data show (line 46).

Suggestion implemented on relevant line from revised manuscript – now line 76.

- 3) Measurements of porosity are made. However the samples may only reveal this porosity under strain – in the negative Poisson’s ratio region. The fact that there is no porosity when unstrained is not conclusive evidence that the mechanism is actually at the molecular level, unless the tests are also made on samples held under strain in the auxetic region.

We thank the reviewer for the suggestion. It was quite a challenge to address this comment as there are almost no reports in the literature of the imaging cross sections of elastic materials *at strain*. To investigate this suggestion we stretched a sample into the auxetic regime and froze it with liquid nitrogen. The sample was then (under cryogenic conditions) removed from the stretching rig, placed into a cryo-SEM shuttle, freeze fractured and studied *via* cryo-SEM. Images from this experiment are found in Fig. 3c and d and. While the surface textures differ in appearance to similar SEM images of the unstrained sample (new images provided in Fig. 2a and b using the same cryo-SEM technique but without straining the sample), it is clear that the sample remains non-porous in the auxetic regime. We have also provided white-light transmission images of a sample being strained through to the auxetic regime. The evident high transparency (& non-scattering) nature of the sample further confirms that no porosity is forming on length scales down to approximately the wavelength of light.

Together the new data and associated discussion confirms our initial assertion that LCE is both non-porous in the unstrained state (Fig. 2), volume conserving under deformation (due to the excellent agreement between curves in Fig 3a), and hence non-porous with strain (as the evolution of pores would require the curves of Fig 3a to differ in shape).

- 4) Line 58 etc and Lines 182 - I have many concerns about the methodology for measuring and calculating Poisson's ratio. The authors make no reference to the literature on this aspect where it is essential to distinguish between small strain and large strain and to use "true strain" at large strains, without reference to the original length.

We thank the reviewer for highlighting that more information is required here. The method for calculating the instantaneous Poisson's ratio was reported by Alderson *et. al.* and Smith *et. al.* (now cited as Refs. 39 and 40) and we have provided details in the methods section (lines 263-304).

- 5) Also, using a constant volume assumption when changes in Poisson's ratio are all about the volume changing is very dubious. This is also related the issue of whether the material is actually highly anisotropic and whether the formulae used are not entirely valid in the anisotropic case.

The updated Fig. 3 and associated discussion (lines 116-136) now make it clear that the LCE is both non-porous and volume conserving. We could identify only one method by which the LCE *could* develop pores and remain, macroscopically volume conserving: simultaneous cavitation and strain-induced-crystallisation both having equal and opposite effects on the sample volume. As neither cavities nor strain-induced-crystals are visible in Fig. 3b, we can rule this unlikely process out and conclude that our material does indeed deform at constant volume.

In terms of anisotropy, it is well known that *monodomain* LCEs are inherently optically and mechanically anisotropic (Refs. 29-32) – indeed we have previously shown this to be true for the presented material (Ref. 30). We carefully checked and confirmed the validity of applying the constant volume condition and equations for calculating the instantaneous Poisson's ratio to the present anisotropic material. In short, as none of the equations require assumptions regarding the anisotropy of the material, they are perfectly valid when applied to our material.

- 6) Whilst there does indeed appear to be a negative Poisson's ratio effect at large strains there needs to be greater clarity on the method of calculation used, reference to earlier works in this area and clarity on the assumptions made about constant volume and anisotropy before the values quoted can be accepted.

We believe that this comment is addressed by our responses to the reviewer's comments 3-5.

- 7) Another important issue is the number of samples tested. I found this difficult to ascertain. Is this simply one sample tested at four different strain rates. Ideally one would wish to see at least 5-10 different samples tested and also some estimate of error bars on the values of Poisson's ratio quoted.

Each test under conditions I-IV (Table 1) and shown in Fig. 4a was performed on a separate sample. Fig. 3a also contains data from yet another sample, allowing us to compare the $\epsilon_x - \epsilon_z$ relationship from direct measurement and from the constant volume condition applied to $\epsilon_x - \epsilon_y$ data. The manuscript thus contains results from 5 separate samples that are representative of the many more samples that have been tested under repeat conditions. We felt that, for clarity of the graphs presented, reporting data from 5 samples is sufficient to demonstrate the repeatability.

Reviewer 2 comments:

- 8) The authors show that in their investigation of a liquid crystal elastomer, it was found that the auxetic behaviour is inherent rather than due to porous structure. The work is not novel, as this has already been taken by Griffin et al 2 decades ago. The specific method employed, however, is new, as does the choice of chemical system.

We would respectfully disagree with the reviewer on the novelty of our work. While the results of the Griffin group “[suggest] the possibility of a negative Poisson’s ratio” (Ref. 27), they importantly were unable to measure a negative Poisson’s ratio for any of their materials. We have updated the introduction to discuss the results of the Griffin group.

- 9) Perusal to figure 3 suggests that the authors adopt the incremental strains, rather than overall strain, for their definition of Poisson's ratio. For example, fig 3a and d show that the strain in z direction is negative throughout the detention in x direction, but fig 3 b and c show that the PR is negative beyond certain range of stretch, which presumably indicated by the upward change in the z strain. As such, the plots of PR in fig 3 b and c should be clarified as "incremental PR".

We thank the reviewer for highlighting the need to clarify this matter. As suggested the graphs and relevant parts of the manuscript have been revised to make reference to the “instantaneous Poisson’s ratio”.

- 10) the quantities l_1 , l_2 , l_3 as well as l_x , l_y , l_z are not clear. May I suggest that these be explicitly displayed and defined on a diagram to prevent any ambiguity and possible misunderstanding that may arise for the readers.

There may have been some typesetting issues when the reviewer’s comments were sent to us so we are not entirely sure which quantities needed explaining in greater detail. We have provided a clearer explanation of l_0 , l_1 , l_2 , l_3 , l_{\parallel} and l_{\perp} and have included Supplementary Data Fig. 4 which clearly shows how the l_x quantities relate to the shape of the LCE polymer conformation.

- 11) where is equation (1)?

Several changes have now been made to the equations presented in the main text and methods and more care has been taken to ensure the equations are labelled appropriately.

- 12) in the first page, the authors missed out some very recent reviews of auxetic systems. These should be incorporated in the early part of the paper for the sake of completeness. The reviews are listed below. X. Ren et al, *Smart Materials and Structures* 27, 023001 (2018), R. Lakes, *Annual Review of Materials Research* 47, 63-81 (2017), T.C. Lim, *Physica Status Solidi RRL* 11, 1600440 (2017)

The updated introduction now includes citations for and aspects from the suggested review articles (and relevant papers within).

Reviewer 3 comments:

- 13) To further demonstrate the hypothesis in the current design concept, chemical functionalization of the polymer backbone or side chain in relation to the increase of r_0 value should have been done. The data presented in current study is based on a single sample tested under different conditions, and the material has been previously reported in *Soft Matter*, 2018,14, 1301-1310. The reviewer feels it is difficult to do comparison and structure-property correlation investigation from different aspects of the materials.
- 14) In addition, to prove the strategy in the material design is versatile, proper physical modifications to achieve an increased magnitude of liquid crystal order parameter so as to obtain a greater negative PR should have been also considered and correlated with underlying mechanism of polymer chain motion in molecular level.

We think that the reviewer has made some excellent suggestions. While we are indeed interested in demonstrating the tuneability of the auxetic behaviour and developing structure-property relations by performing additional studies, the quantity of suggested work goes far beyond the intended scope of this present manuscript. We believe that it would require years' worth of work to study a variety of materials from numerous collaborators to achieve what is suggested.

After a discussion with the editor we have therefore rebalanced the manuscript to truly reflect our intended aim – to report the significant discovery of molecular auxeticity in our LCE. We have left a small comment on how the excellent agreement of our results with WT theory indicates that it should be possible to tune the auxetic behaviour through r_0 . We feel this is an important point to highlight to the readers of our manuscript as it offers a first route to explore structure-property relations associated with molecular auxeticity.

- 15) In addition to the morphology studies by SEM and AFM, as well as relationships between LC order and auxetic behaviour, the authors are suggested to provide more evidence on in-depth study and insightful discussion in relation to the underlying mechanism.

The reviewer's comment highlights the depth of work required to form a picture of the molecular behaviours driving the auxetic behaviour. Our aim here has been to clearly demonstrate the auxetic behaviour of our LCE and to explicitly show that the phenomenon is not the result of a porous geometry. In addition we have been able to use opto-mechanical data and theory to provide a physically reasonable mechanism for the auxetic behaviour based on changes in the anisotropy of the polymer conformation. We would very much like to perform further, deeper analysis of the molecular contributions to the auxetic behaviour but we strongly believe that this is beyond the scope for this first communication.

- 16) Authors are also suggested to demonstrate at least a niche application by using the unique molecular auxetics and its physical properties/functional behaviours.

This discovery is both exciting and novel and we feel that it is most important for this manuscript to focus fully on proving the auxetic behaviour while reporting the fundamental mechanical properties and physical structure of our LCE. The comments from reviewer 1 made the importance of these aspects particularly clear and so we believe that keeping the focus of this manuscript on reporting the discovery of molecular auxeticity to be important. Over the years there have been numerous proposed applications for molecular auxetics which investigators can now explore. Thus, we understand the reviewer's suggestion, but feel that the scope of this paper is clear and that work that will test the proposed applications rigorously should be the focus of future publications.

REVIEWERS' COMMENTS:

Reviewer #2 (Remarks to the Author):

This reviewer thanks the authors for satisfactory revision of their manuscript.

Reviewer #2 was asked to look over the authors' response to reviewer #1's comments. Here are the comments below:

The authors have satisfactorily addressed comments (1) – (4), (6) & (7).

There is a remaining issue pertaining to comment (5), especially in regard to $v_{xy} + v_{xz} = 1$ for v_{xx} traction.

The authors have produced Figure 4 to prove NPR behavior beyond the critical strain, and presumably employed volume constancy to infer v_{xy} , which is not shown anywhere in the manuscript.

For the sake of completeness, the authors should also plot similar graphs as in Figure 4, but with v_y and v_{xy} instead.

By incorporating this new Figure, the more general readers of Nature Communications are able to readily grasp the extent of NPR in the LCE on both xz and xy planes.

Reviewer #3 (Remarks to the Author):

The concerns have been well addressed in the revised manuscript. The reviewer feels that the article can be accepted for publication.

REVIEWERS' COMMENTS:

Reviewer #2 (Remarks to the Author):

This reviewer thanks the authors for satisfactory revision of their manuscript.

Reviewer #2 was asked to look over the authors' response to reviewer #1's comments. Here are the comments below:

The authors have satisfactorily addressed comments (1) – (4), (6) & (7).

There is a remaining issue pertaining to comment (5), especially in regard to $v_{xy} + v_{xz} = 1$ for v_{xx} traction.

The authors have produced Figure 4 to prove NPR behavior beyond the critical strain, and presumably employed volume constancy to infer v_{xy} , which is not shown anywhere in the manuscript.

The above response to the editor's comments also address this reviewer comment. We believe that we had discussed the use of the constant volume condition alongside figure 3 and figure 4. However, we appreciate that mentioning the constant volume condition earlier in the text (as we have now done) makes the methodology clearer and the figures easier to understand. Full details of how the constant volume condition is employed is also given in the relevant methods section where we feel it is most appropriately placed. We have chosen to leave these details in the methods section as not to disrupt the flow of the text in the results section.

For the sake of completeness, the authors should also plot similar graphs as in Figure 4, but with v_y and v_{xy} instead. By incorporating this new Figure, the more general readers of Nature Communications are able to readily grasp the extent of NPR in the LCE on both xz and xy planes.

As suggested the relevant figures have been moved from the supplementary information to figure 4. We felt that an equivalent figure figure 4c for the xy plane should not be produced as the concept of ϵ_c pertains only to the critical strain for the emergence of auxetic behaviour in the z direction and has not significance to strains along the y axis.

Reviewer #3 (Remarks to the Author):

The concerns have been well addressed in the revised manuscript. The reviewer feels that the article can be accepted for publication.